# Zebrafish—A Suitable Model for Rapid Translation of Effective Therapies for Pediatric Cancers

**DOI:** 10.3390/cancers16071361

**Published:** 2024-03-30

**Authors:** Debasish Roy, Bavani Subramaniam, Wai Chin Chong, Miriam Bornhorst, Roger J. Packer, Javad Nazarian

**Affiliations:** 1Center for Genetic Medicine Research, Children’s National Hospital, Washington, DC 20012, USA; droy@childrensnational.org (D.R.);; 2DIPG/DMG Research Center Zurich, Children’s Research Center, Department of Pediatrics, University Children’s Hospital Zürich, 8032 Zurich, Switzerland

**Keywords:** pediatric cancers, zebrafish, brain tumors, drug toxicity, targeted therapy, animal model

## Abstract

**Simple Summary:**

Cancer stands as the leading cause of death among children and adolescents in the United States. Some of these pediatric cancers are highly aggressive and complex in nature. Current in vitro models often fail to accurately replicate the tumor microenvironment, while in vivo models face time and cost constraints. This review article emphasizes the unique advantages of zebrafish models in pediatric cancer research due to their genetic similarity to humans, short experimental timeline, ease of genetic manipulations, rapid in vivo tumor development, and transparent bodies that facilitate precise tumor cell tracking at single-cell resolution. Through a comprehensive analysis of existing literature and experimental findings, the article highlights the potential of zebrafish as a valuable preclinical model for studying tumor biology, expediting drug discovery and screening processes, and implementing personalized medicine strategies for treating pediatric cancers.

**Abstract:**

Pediatric cancers are the leading cause of disease-related deaths in children and adolescents. Most of these tumors are difficult to treat and have poor overall survival. Concerns have also been raised about drug toxicity and long-term detrimental side effects of therapies. In this review, we discuss the advantages and unique attributes of zebrafish as pediatric cancer models and their importance in targeted drug discovery and toxicity assays. We have also placed a special focus on zebrafish models of pediatric brain cancers—the most common and difficult solid tumor to treat.

## 1. Introduction

Cancer is the leading cause of death by disease in children and adolescents in the United States, with an incidence rate of approximately 175 per 1 million [1]. Leukemia constitutes 28% of all childhood cancers, while all solid tumors make up approximately 40% of childhood cancers [2]. Among solid tumors, tumors involving the central nervous system (CNS), including the brain and/or spine, are the most common solid tumors in this population [1]. Although many patients with solid tumors are successfully treated with multimodal therapy (chemotherapy, radiation therapy, and surgery), other tumors including relapsed/recurrent sarcomas, malignant melanomas, stage four neuroblastoma, and malignant brain tumors have poor overall survival [3,4]. In addition, treatment often results in long-term side effects and impacts quality of life. The recent understanding of tumor etiology has provided an opportunity to establish biology-informed clinical interventions. Through advanced sequencing techniques and large cooperative studies, new therapeutic approaches that target molecular alterations specific to the tumor and the tumor environment are being developed. These therapies will likely be critical for successful management and long-term survival for many solid tumors in the future.

Preclinical efficacy assays and safety studies are an essential component of novel therapy development, as this allows researchers to study both on-target and off-target effects of novel agents and provides information about potential toxicity and dose requirements for clinical trial development [5]. Pre-clinical models that are easy to use, relatively inexpensive, timely, and representative of the tumor and tumor micro-environment are most informative. In recent times, zebrafish have acquired huge popularity as a reliable model for cancer research due to their genetic similarities with humans, ease of genetic manipulation, in-vivo monitoring of tumor progression due to their transparent body structure in early developmental stages, high fecundity, low maintenance cost, and the ability to conduct drug and chemical screening in large numbers of animals. In this article, we review the potential and the unique attributes of zebrafish as pediatric cancer research models, as well as their comparative advantages over other preclinical models.

## 2. In Vitro Models for Pediatric Cancers

Cancer cell lines are the most applicable and cost-effective models for studying a tumor’s molecular characteristics as well as high throughput screening for the identification of potential drug targets. Cancer cells that are predominantly used for investigating tumor biology and in vitro drug testing fall into two categories: (i) cell lines developed from human samples and (ii) cell lines derived from animal models. These two categories are further subdivided into (i) primary tumor cells, which are derived directly from the tumor samples; (ii) genetically altered healthy cells to convert them into converted to tumor cells; and (iii) iPSC-derived tumor cells [6,7,8,9,10]. These cell lines normally retain their tumorigenic characteristics and are often immortalized to obtain an indefinite life span. Most of the cancer cell lines are simple to create, can be stored for decades, and are useful in studying the molecular mechanisms of tumor development and metastasis [11,12].

The major drawbacks of the in vitro cell culture models are their inability to provide a proper tumor microenvironment and failure to address the cause of cancer recurrence [13,14]. The tumor microenvironment is made up of tumor cells; stromal cells such as endothelial cells, immune cells, and fibroblasts; and extra-cellular matrix (ECM) components, like collagen, laminin, fibronectin, etc. [15,16,17]. These tumor–stromal–ECM interactions cannot be captured in traditional monolayer cell culture. Recent advancements in molecular cell biology and tissue engineering have resulted in the development of more robust ‘3D cell culture’ and ‘organoid’ models to address these shortcomings. In 3D cell culture, cells are allowed to grow and interact with the surrounding ECM framework in three dimensions. Organoids, which are also developed in a 3D cultural framework, are miniature versions of organs derived from stem cells that replicate the morphology, cell types, and functions of their in vivo counterparts [18,19,20]. Organoids can provide tumor microenvironment and cellular interactions to a certain extent, which makes them more efficient for drug screening than conventional cell culture models [21]. In 2020, Krieger et al. developed a cerebral organoid model that served as a scaffold for glioblastoma (GBM) cell invasion. Transcriptional analysis of GBM organoids revealed possible ligand–receptor interactions between tumor and organoid cells [22]. Huang et al. developed an organoid model for meningioma, and immunohistochemistry analysis revealed striking similarities in cellular heterogeneity between meningioma patient samples and meningioma organoids, which include tumor cells, T-lymphocytes, macrophages, and vascular endothelial cells [23]. However, high-cost, complex model development protocols, lack of proper vasculature, and poor controllability limit the applications and clinical implications of these models [24]. 

## 3. In Vivo Models for Pediatric Cancers

A number of animal models have been established for studying pediatric tumors over the past six decades. Invertebrates, including *C. elegans* and *Drosophila*, to vertebrates, such as rodents, felines, canines, and non-human primates [25,26,27,28,29], have been utilized to evaluate potential therapeutic strategies and cures for this deadly disease. Rodent models, particularly mice and rats, are the most used animal models in pediatric cancer research. Having more than 80% genetic similarity to humans as well as comparable anatomical, molecular, and biological characteristics make them the most popular models for human disease research [30,31]. Mouse tumor models, the most popular model among cancer biologists, can be broadly divided into two categories: (i) xenograft models, and (ii) transgenic or genetically engineered mouse models (GEMMs). Traditional patient-derived xenograft (PDX) models are created by implanting or injecting human malignant cells into immunodeficient mice, as wild-type mice with functional immune systems reject foreign tumor cells or tissues. By contrast, patient-derived orthotopic xenografts (PDOXs) are created by implanting tumor tissue or cells from a patient into a mouse in the same anatomic location in order to further emulate the tumor microenvironment. The major disadvantages of rodent models are the differences in physiological parameters compared with humans, the lower complexity of the tumor microenvironment, and the rapid clearance of drugs from the body due to fast liver metabolism [32,33,34,35,36]. In addition to that, they are also time-consuming and expensive to develop and cannot be shared among scientists, and perhaps most significantly, they require immunocompromised animals for developing patient-derived xenograft models [37,38]. Although scientists have created immunocompetent syngeneic mouse models (SMMs) and engrafted tumor cell lines from the same genetic background to avoid immune rejection, they lack inter- and intra-patient tumor microenvironment heterogeneity and differ in tumor growth kinetics [39,40]. GEMMs, on the other hand, are developed to study the role of specific genes by deleting, overexpressing, or mutating, which results in spontaneous tumor formation. Although they are the most accurate histological and genetic models of pediatric cancers, they have several drawbacks, including the variable and unpredictable nature of tumor development, a lack of complex genomic landscapes found in human tumors, and a longer tumor development time [41,42]. Zebrafish pediatric cancer models bridge this gap. In the following sections, we will detail the benefits of zebrafish models and highlight shortcomings that need to be addressed in future studies. A comparison between different in vitro and in vivo models is summarized in Table 1.

## 4. Zebrafish—A Tiny Human

Zebrafish (*Danio rerio*) were first discovered by F. Hamilton in the Ganges River in northeastern India in the 1820s and were described as “beautiful fish” with “several blue and silver stripes on each side” [43,44]. Zebrafish are tropical, freshwater fish belonging to the minnow family (Cyprinidae) of the order Cypriniformes. It is an ideal organism to maintain under laboratory conditions due to its small size, high fecundity rate, and ability to reach sexual maturity by 3 months of age [45]. Zebrafish breed throughout the year and produce a large number of embryos per cross. Furthermore, fertilization is external, and developing embryos are transparent, allowing the visualization of almost all organs using a simple dissection microscope [46]. The embryo develops into a free-swimming larva 3–4 days post fertilization (dpf). By observing these advantages, Dr. George Streisinger at the University of Oregon realized the potential of zebrafish as a suitable model organism and introduced them as a model system in biological research in 1972. Sanger’s Institute initiated the Zebrafish Reference Genome Sequencing project in 2001, which was completed in 2013 [47]. The sequencing data revealed 71.4% of the human genome to be conserved in zebrafish, with 82% of human disease-related genes having at least one ortholog in zebrafish [47]. In 2020, Yang et al. developed a detailed map of zebrafish transcriptomes, cis-regulatory elements, heterochromatin regions, methylomes, and 3D genome organization using a combination of advanced techniques, including RNA sequencing, assay for transposase-accessible chromatin using sequencing (ATAC-seq), chromatin immunoprecipitation sequencing (ChIP-Seq), whole-genome bisulfite sequencing (WGBS), and chromosome conformation capture (Hi-C) experiments. When comparing zebrafish regulatory elements with those of humans and mice, they found both evolutionarily conserved and species-specific regulatory sequences and networks [48]. 

Zebrafish and humans share conserved organ systems, including the mouth, eyes, brain and spinal cord, intestine, pancreas, liver with bile ducts, kidney, esophagus, heart, ear, nose, muscle, blood, bone, cartilage, and teeth (Figure 1). Most of the important biological and metabolic pathways are also similar in zebrafish (KEGG pathway: zebrafish) providing the opportunity to study an array of human diseases using zebrafish models.

## 5. Zebrafish Models for Human Disorders

Since the 1980s, zebrafish have been employed in biomedical research to model a broad spectrum of human diseases. Zebrafish models have been successfully developed to study disorders ranging from neurodevelopmental disorders to those that are metabolic in nature [49,50]. 

Zebrafish models demonstrated comparable and quantitative changes in their social and cognitive behavior, altered locomotion, and even increased head size, all of which are typical of autism spectrum disorders (ASDs) [49,51,52,53]. Similarly, another neurodevelopmental disorder, Rett syndrome, was effectively modeled in zebrafish, which had observable behavioral alterations throughout their early infancy, including spontaneous and sensory-evoked motor abnormalities as well as thigmotaxis deficits [54,55]. Zebrafish have also been effectively utilized to model neurodegenerative diseases [56], behavioral disorders [57,58], neuromuscular disorders [59,60,61,62,63], aging-related disorders [64,65,66], hematopoietic disorders [67,68,69,70], renal diseases [71,72,73,74], liver diseases [74,75,76,77], and eye diseases [78,79,80,81], to mention a few. In recent years, zebrafish have also been used to study infectious viral diseases, such as Zika [82] and COVID-19 [83].

## 6. Zebrafish Models for Cancer Research

The utilization of zebrafish in cancer research has a long history. It was first reported in 1965 when Dr. Stantion induced irreversible liver damage and hepatic neoplasms in zebrafish by using the water-soluble carcinogen diethylnitrosamine [84]. Cancer modeling in zebrafish is mostly achieved in three ways: (i) the forward genetics method, (ii) the reverse genetics approach, and (iii) xenotransplantation. In 1996, Haffter et al. published their landmark study of a large-scale mutagenesis screen, generating numerous mutant zebrafish lines that are still useful in present-day cancer research [85]. Utilizing a similar forward genetics approach, Lee G. Beckwith and Dr. Jan M. Spitsbergen developed many neoplasms, such as papilloma, hemangiomas, hepatocellular adenoma, and rhabdomyosarcoma, by using common mutagens, like ethylnitrosourea (ENU) and N-methyl-nitrosoguanidine (MNNG) [86,87]. Several innovative strategies for gene knockdown, gene editing, and transgene insertion into the zebrafish genome have emerged in recent years. These reverse genetic techniques attempt to create a loss-of-function phenotype or transfer genes identified to be altered in human patients with cancer into fish. 

Mutations in cancer predisposition genes (CPGs) are often associated with an increased susceptibility to cancer. Inherited mutations in CPGs cause approximately 10% of cancers in humans. More than 100 CPGs with diverse cellular and molecular functions have been identified, which might provide insight into the prevention, diagnosis, and optimization of cancer management [88,89]. Mutation CPGs linked to DNA repair, genome stability, signaling pathways, transcriptional regulation, epigenetic modifications, telomere maintenance, and metabolism often increase the susceptibility to tumor development. The utilization of zebrafish models to investigate the roles of CPGs in cancer was extensively discussed in a review by Kobar et al. [90].

## 7. Pediatric Cancer Models in Zebrafish

Compared with adult tumors, pediatric tumors often exhibit low mutational burdens and are frequently driven by singular driver genes, oncoproteins, or copy number variations [91]. The most common types of cancers diagnosed in children and adolescents (ages 0–19 years) are leukemias, CNS tumors, and sarcomas [92]. In the subsequent sections, we elaborate on non-central nervous system (CNS) and CNS pediatric cancer models in zebrafish.

## 8. Non-CNS Pediatric Cancer Models in Zebrafish

The most commonly diagnosed cancer in children is acute lymphoblastic leukemia (ALL). Although the survival rate for this type of cancer is high, many long-term detrimental side effects are seen in these patients due to the current treatment procedures [93]. Given these considerations, ongoing studies aim to minimize treatment-related toxicity and develop targeted therapies for recurrence and high-risk patients. ALL can be broadly classified into B-cell acute lymphoblastic leukemia (B-ALL), and T-cell acute lymphoblastic leukemia (T-ALL). Mariotto et al. developed a xenograft model for B-ALL in zebrafish. They identified BCL2-associated athanogene-1 (BAG1) as a potential target due to its increased expression during cancer relapse. They conducted transient knockdown of BAG1 protein expression in RS4;11 leukemia cells and xenografted them into zebrafish embryos two days post fertilization (dpf). Their study revealed that anti-cancer drugs such as dexamethasone, daunorubicin, and the BCL2 inhibitor ABT-737 exhibited greater sensitivity in BAG1-depleted cells without any toxicity, whereas pan-BCL inhibitors caused cytotoxic effects in zebrafish [94]. T-ALLs are biologically different from B-ALLs, and they can be subgrouped according to targetable pathways, such as Notch, Jak/Stat, PI3K/Akt/mTOR, and MAPK [95]. Hedgehog pathway mutations, particularly those affecting *PTCH1* expressions, are quite common (16%) in patients with T-cell acute lymphoblastic leukemia (T-ALL). Burns et al. developed a *ptch1* CRISPR knockout zebrafish model for T-ALL. Their findings indicated that *ptch1* mutations expedited the onset of *notch1*-induced T-ALL. Additionally, the study suggested that the inhibition of the Hedgehog pathway could serve as a targeted therapy for high-risk T-ALL [96].

Rhabdomyosarcoma, which is the most common soft-tissue sarcoma diagnosed in children, constitutes approximately 3–4% of all pediatric cancers [97,98]. Alterations in the RAS/MAPK signaling pathway are reported in many patients with rhabdomyosarcoma [99] Kahsay et al. demonstrated a significant downregulation of the RAS/MAPK pathway in *pax3* double mutant zebrafish (pax3a^−/−^; pax3b^−/−^), which resulted in a delayed progression of kRAS-induced rhabdomyosarcoma [100].

While dysregulation of HES3 was observed in children with fusion-positive rhabdomyosarcoma, the precise mechanism of HES3 involvement in this pediatric cancer remained unclear. To elucidate this, Kent et al. developed a *her3* (ortholog of human *HES3)* knockout zebrafish model of rhabdomyosarcoma. Transcriptomic analysis of the *her3* mutant zebrafish unveiled the impact on several cancer-related gene pathways, along with the downregulation of genes involved in organ development, such as *pctp* and *grinab* [101].

We have listed examples of non-CNS pediatric cancer models in zebrafish in Table 2. 

## 9. Pediatric CNS Cancer Models in Zebrafish

CNS tumors are the second-most common tumors and the most common solid tumors in children and the leading cause of pediatric cancer-related death [1,134,135]. Alongside various in-vitro and in vivo models, many zebrafish models have also been developed to study these deadly pediatric cancers. Before discussing different zebrafish brain cancer models, we believe it is important to emphasize the remarkable similarities between zebrafish and human brain anatomy and functions. 

## 10. Zebrafish Brain 

Although the zebrafish is a teleost, the general macro-organization of the brain and cellular anatomy of zebrafish and humans are remarkably similar (Figure 2). The zebrafish brain has all the key brain anatomical components seen in humans [136]. Adult zebrafish have well-developed forebrain, midbrain, and hindbrain with prominent optic tectum, thalamic, and hypothalamic regions [137]. The embryonic forebrain of zebrafish develops into the telencephalon, diencephalon, hypothalamus, and retina [138,139]. The zebrafish telencephalon comprises the pallium, sub-pallium, and olfactory bulb, and it is important for their social behavior, memory, and emotions [140,141]. The thalamus, pineal body, and habenula constitute the zebrafish diencephalon, which regulates their attention, alertness, and circadian patterns [142]. The tectum and tegmentum are two key structures in the zebrafish midbrain that are important for vision, hearing, motivation, and reward [143,144]. The zebrafish hindbrain appears as a distinct structure posterior to the midbrain at embryonic developmental stages. It is separated from the midbrain by a temporary structure called the midbrain–hindbrain border (MHB), which is absent in adult zebrafish. The hindbrain regulates eye, jaw, and head movement and gives rise to the cerebellum, which is responsible for motor control, sensory input reception and response, cognition, emotion, and learning [145,146,147,148,149]. Human and zebrafish developmental gene expression shifting patterns are also remarkably comparable, highlighting the significance of this model in studying childhood brain disorders [150]. The neurochemical aspects of the zebrafish brain are also very similar to those of humans. Zebrafish possess all major neuromediators, including receptors, neurotransmitters, transporters, and enzymes [151,152,153,154,155,156,157,158,159]. Zebrafish have well-developed functional neuroendocrine systems like those present in mammals. The zebrafish stress response is driven by the hypothalamic-pituitary hormonal cascade and is regulated by cortisol, which acts through the glucocorticoid receptor, which is very similar in humans [160,161,162,163]. 

## 11. Zebrafish Models of Childhood CNS Cancer

Gliomas are the most frequent pediatric tumors, accounting for around half of all brain cancer cases recorded in children [134,135]. These CNS tumors can be classified as low- or high-grade gliomas depending on their malignant nature. Zebrafish glioma models, like other in vitro and in vivo models, have been effectively utilized for investigating these lethal CNS tumors, drug toxicity, and novel therapeutic inventions. Among Glioblastomas, the *NF1* gene was found to be mutated in 20% of these cases. Shin et al. developed a glioblastoma model by developing a stable mutant *nf1* zebrafish line using zinc finger nuclease (ZFN) and targeting induced local lesions in genomes (TILLING) gene editing techniques. They reported that the zebrafish carried at least one copy of the type *nf1* allele (*nf1a* or *nf1b*), were viable and fertile, and showed no tumor formation. However, when both the alleles of *nf1a* and *nf1b* genes were absent, the nf1a^–/–^; nf1b^–/–^ larvae displayed melanophore defects 6 days post fertilization (dpf), and they did not survive beyond 10 dpf. Although these nf1a^–/–^; nf1b^–/–^ larvae did not develop tumors, they exhibited defects in glial cells and pigment cells, which are often observed in *NF1* human patients. Moreover, in a *p53* mutant background, *nf1a^+/–^*; *nf1b^–/–^*; *p53^e7/e7^* zebrafish developed adult-onset high-grade gliomas and malignant peripheral nerve sheath tumors, which resemble human *NF1* HGGs [165].

The tumor microenvironment is critical for tumor cell survival and metastasis, and microglia and infiltrating macrophages are two major cell types found in about 30% of HGG tumor tissues [166]. Chia et al. reported in 2018 that the neuron-specific overexpression of human AKT1 in zebrafish resulted in a large increase in macrophage and microglia populations. They developed the transgenic zebrafish lines *Tg*(NBT:∆LexPR-lexOP-pA; *mpeg1*:EGFP) and *Tg*(mpeg1:mCherry; p2ry12:p2ry12-GFP) and also utilized cxcr4b^−/−^ mutant zebrafish to study the macrophage and microglial cell behavior inside the developing zebrafish brain. They demonstrated that the peripheral macrophage infiltration into the brain occurred through Sdf1b–Cxcr4b signaling [167].

CNS primitive neuro-ectodermal tumors (CNS-PNETs), presently classified as embryonal tumors, belong to the embryonal family of malignant childhood brain tumors. CNS primitive neuro-ectodermal tumors (CNS-PNETs) constitute a heterogeneous group of brain embryonal tumors that includes all CNS embryonal malignancies not diagnosed as medulloblastoma, atypical teratoid/rhabdoid tumors (AT/RT), or embryonal tumors with multilayered rosettes (ETMRs). Histologically, CNS-PNETs are identified by the presence of small, poorly differentiated cells and a mixed population of both glial and neuronal lineages. The 5-year progression-free survival rate for CNS-PNETs is currently as low as 30% [168,169,170,171]. Schultz et al. created an embryonal brain tumor model with homozygous *rb1* loss of function via CRISPR mutagenesis that closely mirrored human CNS-PNETs [172]. Zebrafish *rb1* tumors, like the human oligoneural OLIG2+/SOX10+ CNS-PNET subtype, displayed an over-expression of neural progenitor transcription factors, such as *sox8b*, *sox10*, and *olig2*, as well as the *erbb3a* oncogene [172]. Another zebrafish embryonal tumor model was developed by Modzelewska et al. through NRAS activation in Olig2+/Sox10+ oligoneural precursor cells. Genomic analysis revealed a striking similarity with human NB-FOXR2 CNS-PNET subgroup tumors. They also generated an orthotropic embryonic brain tumor transplantation assay for drug screening by allografting tumors from *Tg (mitfa^w2^*; *p53^M214K^*; *Tg(sox10:mCherry-NRAS^WT^) fish into 2dpf mitf^w2^* embryos and found that MEK inhibitors could remove these tumors in 79% of the fish [173].

Apart from transgenic tumor models, there are many excellent studies in which the investigators used zebrafish xenograft models (both tumor cell lines and patient-derived tumor cells) to study tumor biology, drug screening, and future immunotherapeutics. Zebrafish have a delayed adaptive immune system, as T and NK cells develop at 5 dpf and B cells by 21 dpf, which provides researchers with a short but valuable window to perform xenograft experiments [174]. These xenograft models are fast and can be developed only in 5 days, and most importantly, they match mouse xenograft models in terms of tumor growth and initiation [175]. Zebrafish xenograft models can be used for studying tumor biology or for drug discovery purposes. Hamilton et al. developed a xenograft model by injecting U87 and U251 glioblastoma cells to investigate the interaction between microglia and glioma in vivo. Their findings revealed variations in the growth rates and microglial interactions of different glioma cells. Notably, when U87 cells were xenografted into *irf8*^−/−^ zebrafish mutant embryos, which lack microglia, the injected tumor cells exhibited a significant reduction in cell survival compared with wild-type embryos. This study underscores the crucial role of microglia in the dynamics of glioma growth [176]. Pudelko et al. orthotopically injected patient-derived tumor cells in zebrafish larvae to study the effect of MTH1 inhibitors in glioblastoma in real-time. They found that their in-house-developed MTH1 inhibitor, TH1579, was able to successfully irradicate glioblastoma stem stems in zebrafish [177]. 

Most xenograft experiments in zebrafish are performed between 5–7 days to avoid immune rejection. To address this issue, Prof. David Langenau’s group at Massachusetts General Hospital and the Harvard Stem Cell Institute developed an immunodeficient (*prkdc^−/−^*, *il2rga^−/−^) casper*-strain zebrafish that lacked T, B, and natural killer (NK) cells. These immunodeficient zebrafish can survive at 37 °C, which is ideal for tumor cell growth, and they can be engrafted with up to 1.5 × 10^6^ tumor cells. The xenografted cells were able to survive in adult zebrafish for 28 days [178,179]. These fish models can be utilized in long-term xenograft studies and studies on accurate preclinical drug dosing in adult zebrafish [180]. We have listed the most common childhood brain cancer models developed in zebrafish in Table 3.

## 12. Humanized Zebrafish

Although xenograft and transgenic models are excellent tools for studying human illnesses because they provide a better microenvironment, none of them can mimic one hundred percent of human conditions. To bridge this gap, scientists are continuously developing humanized model organisms. Zhu et.al. published an excellent review on humanized mouse models and their benefits in studying human health and disease [185]. Similar to mice, humanized zebrafish models have been created to better understand human disorders and bridge the gap between species differences. The first such humanized zebrafish model was developed by Rajan et. al. in 2020, who created ‘GSS fish’ that expressed human hematopoietic-specific cytokines, GM-CSF, SCF, and SDF1α. By stimulating the self-renewal and multilineage differentiation of human hematopoietic stem and progenitor cells, these GSS zebrafish could establish a better microenvironment for human leukemia cells [186]. Similarly, another humanized zebrafish model was developed by Häberlein et. al. to study multiple sclerosis. They first created a zebrafish *gpr17* loss-of-function transgenic animal and inserted the human *GPR17* gene into their genome so that they could only express human *GPR17 protein*. This humanized zebrafish model could be essential in studying multiple sclerosis and identifying pro-remyelination compounds [187]. Many more humanized zebrafish will be generated in the near future to identify the optimal treatment for various human diseases at a personalized level.

## 13. Zebrafish as a Tool for Cancer Drug Discovery

The zebrafish model serves as a link between in vitro and in vivo investigations in mammals. This model is powerful in terms of its range of applicability and research tractability. In past decades, numerous studies have used zebrafish models to understand disease biology, drug efficacy, and toxicity. In cancer research, drug toxicity is a major concern. When compared with in vitro and other in vivo models, drug screening and toxicity studies in zebrafish can be performed in a comparatively short period of time and with more physiological relevance. Patient-derived avatar models in early larval stages in zebrafish are ideal for large-scale drug screening purposes. Xenografted embryos can be maintained in 96-well plates, and they easily absorb water-soluble drugs from the culture media, which is mostly E3 medium or fish water [188]. Drugs that are non-toxic and effective against tumor cells are selected for future studies, which include target validations, in vivo murine models, and, finally, translation to clinical trials (Figure 3). These drug toxicity tests are often performed in accordance with OECD recommendations, which are compilations of the most relevant internationally agreed-upon testing methodologies used by governments, industry, and independent laboratories to assess the safety of chemicals (https://www.oecd.org (accessed on 1 May 2023)). 

To measure toxicity in zebrafish embryos, specific endpoints, such as egg coagulation, lack of somite development, non-detachment of the tail, and the absence of heartbeat, are used. In addition to these endpoints, researchers frequently quantify the heartbeat, coiling behavior, hatching percentage, yolk sack edema, pericardial edema, yolk sac necrosis, and tail curvature to measure the toxicity level of any compound (Figure 4). These additional observations are important for assessing the maximum tolerated dose/concentration. 

## 14. Zebrafish: A Rapid and Cost-Effective Pediatric Cancer Model Organism

Time is essential in pediatric cancer treatment. The overall survival time of patients with pediatric cancer varies depending on the cancer type. CNS tumors, particularly aggressive ones like atypical teratoid tumors, exhibit a remarkably low overall survival time of just 6 months, whereas in patients with diffuse intrinsic pontine gliomas (DIPGs), the survival rate is less than one year [189,190]. Developing xenograft models in rodents to study these cancers takes at least 3–6 months [191]. Moreover, conducting large-scale drug screenings in rodent models is not only time-consuming but also presents ethical concerns and substantial costs. On the other hand, zebrafish models offer a cost-effective and rapid in vivo platform for studying brain cancers. The estimated cost for the in vivo screening of a single drug in zebrafish is approximately USD 300, making it 500 times more economical than comparable studies in rat models. In a parallel two-week in vivo study comparing zebrafish and mice, zebrafish models were found to be five times more cost-effective than their mouse counterparts [45]. Most of the xenograft studies in zebrafish can be completed within a week and yield results comparable to those of their mouse xenograft counterparts [175]. 

## 15. Concluding Remarks and Future Perspectives

Although high-grade childhood cancers are often termed as ‘rare’ diseases, they are a major concern across the medical field because of their severity and social impact. Aggressive pediatric cancers are also different from adult tumors in terms of age of onset, occurrence, progression, and treatment. Most of the time, patients’ mean survival time is far shorter, and it is difficult or impossible to develop rodent PDX models in that timeframe. In recent decades, the zebrafish has become an important model organism in strengthening our understanding of cancer etiologies, the involvement of the immune system in the cancer microenvironment, and drug screening. Whereas transgenic zebrafish models can be used to study tumor biology, disease progression, and cellular interactions in detail, xenograft models can be created rapidly and utilized to test hundreds of medicines, and most importantly, they can be used to find new therapies on a personalized level. Recent advances in zebrafish research include ‘humanized zebrafish’, which allows for a more intimate tumor microenvironment, and ‘immunodeficient zebrafish’, which allows for long-term xenograft trials, making this model more relevant and attractive. 

There are some disadvantages of using zebrafish as a model for toxicity and drug testing. In the embryo toxicity assay, commonly, drugs or compounds are directly added to the water, and zebrafish embryos are exposed to these solutions. In the early stages of development, the protective chorion of zebrafish embryos may prevent the entry of compounds that have molecular weights of more than 4000 Da [192]. Additionally, the zebrafish skin also acts as a barrier to many drugs. In both low-water-soluble and water-insoluble drug testing, low sensitivity and inconsistent results are common outcomes. Various factors, including different routes of drug exposure, inaccuracies in determining the drug concentration within the embryos, temperature variations, lack of adaptive immunity in larvae, and external environmental conditions, can collectively reduce the clinical relevance of the obtained results. Consequently, alternative administration routes, such as gavaging and microinjection, become necessary, leading to time-consuming and technically challenging procedures. Addressing these challenges requires a deep understanding of the drug/compound’s structure, size, and solubility as well as careful experimental design and the implementation of various administration methods.

In the future, many unanswered questions need to be addressed. The refinement of existing models, drug delivery methods, and the development of more immunodeficient and humanized zebrafish models will help us solve many unresolved questions related to childhood cancers. Zebrafish have already demonstrated their relevance in several human disease studies, including studies on pediatric cancers, and we hope that they will soon be acknowledged as a model animal for clinical trials.

## Figures and Tables

**Figure 1 cancers-16-01361-f001:**
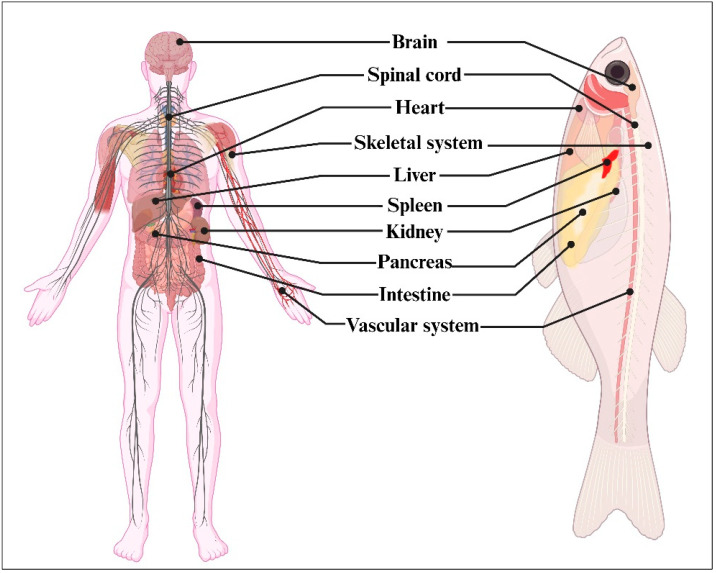
Depiction of conserved organ systems in humans and zebrafish.

**Figure 2 cancers-16-01361-f002:**
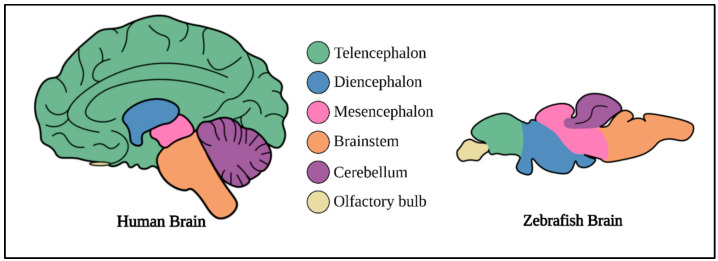
Comparison of anatomy between human and zebrafish brain. Image is adapted from the review article by Haynes et al., 2022 [164].

**Figure 3 cancers-16-01361-f003:**
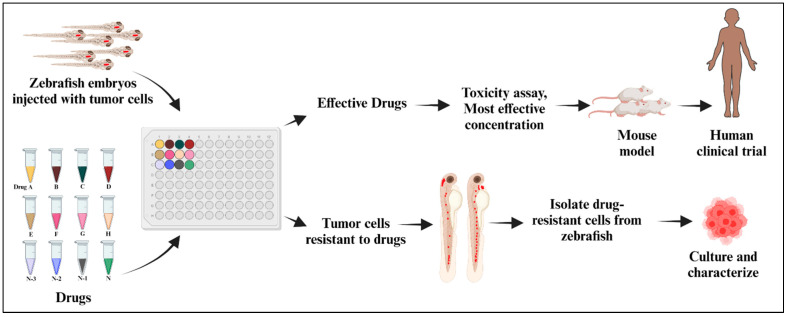
Anti-cancer drug screening using zebrafish xenograft model.

**Figure 4 cancers-16-01361-f004:**
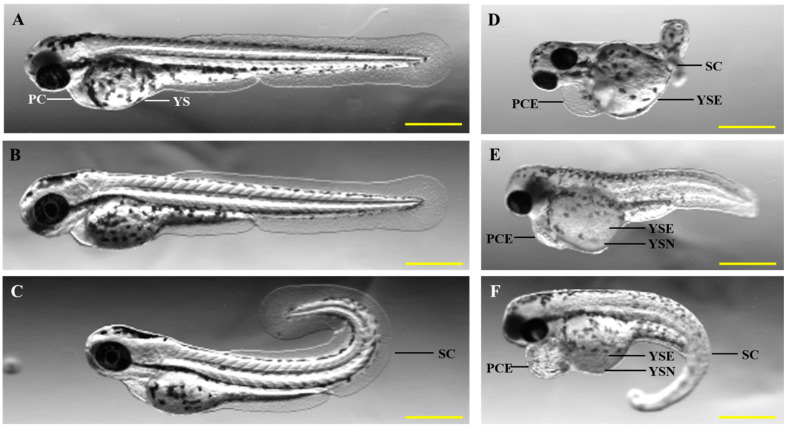
Developmental defects found in drug toxicity. (**A**) Wild-type control 4 dpf embryo. (**B**–**F**) Different developmental defects in 4 dpf embryos due to drug toxicity. PC—pericardium, YS—yolk sac, PCE—pericardial edema, YSE—yolk sac edema, SC—spinal curvature, YSN—yolk sac necrosis. Scale bar = 1.5 mm.

**Table 1 cancers-16-01361-t001:** Comparison between zebrafish and other childhood cancer models.

Model	Cost	Advantages	Disadvantages	Drug Screening Throughput
Cell culture	Low	Immortality, rapid growth, robustness, ease of genetic modifications, maintenance, and storage	Long-term culture can result in the development of cells that are genetically distinct from primary tumor cells. There are no tumor microenvironments.	Very High
Organoids	High	Similar tumor heterogeneity, characteristics, and microenvironments to human systems.	Technically difficult to develop, costly, and variable in growth.	High
Rodents	Very High	Replicate tumor microenvironment, genetic alterations, and pharmacodynamics as closely as possible to humans.	Time-consuming; lack immune interactions. PDX models primarily depend on tissue integrity; highly variable in nature.	Low
Zebrafish	Intermediate	External fertilization, large number of offspring, ease of transplantation, high efficiency in genetic manipulation, rapid tumor engraftment, and the development of tumors with histopathology similar to that of humans. Cell tracking in embryos and larvae is very easy due to their transparent bodies.	Difficulties in accurately measuring drug concentration in larval plasma, hindering drug absorption, distribution, metabolism, and excretion (ADME) studies, along with a lack of adaptive immune system in larvae, restrict the direct prediction of clinical dosage from zebrafish to humans.	High
Drosophila	Low	Short lifespan, large number of progenies, ease of genetic manipulation, drug screening, dissection of complex tissues.	Tumors can grow for a limited time and thus are not suitable for long-term studies. Angiogenesis and adaptive immunity cannot be studied. A few organ-specific tumors, such as pancreas, liver, and lung tumors, cannot be studied due to a lack of homologous organs.	High
*C. elegans*	Low	Conserve signaling pathways, genetic similarities, ease of genetic manipulation, transparent body.	They do not develop cancers in a similar way to humans. They lack adaptive immunity, angiogenesis, and organ systems comparable to those of humans.	High

**Table 2 cancers-16-01361-t002:** Pediatric non-CNS cancer models in zebrafish.

Cancer	Studied Gene/Cell Line/Drug	Method	Zebrafish Line Used/Developed	Reference
Acute lymphoid leukemia	*BAG1*	RS4;11 cell xenograft	*Tg(fli1:GFP)*	[94]
	*Ext1* and *Ext 2*	Morpholino-based knockdown	*Tg(Tp1bglob:eGFP)*	[102]
	*MYC*	Transgenic	*Tg(rag2:hMYC)*, *Tg(lck:eGFP)*, *Tg(hMYC;GFP)*	[103]
Adenoid Cystic Carcinoma	*MYB*	Transgenic	Zebrafish blastomeres, *Tg(c-myb-GFP)*, *Tg(c-myb:GFP; lyz:dsRed)*, *Tg(c-myb:GFP; mpeg1:mCherry)*	[104]
Chronic Myeloid Leukemia	*BCR/ABL1*	Transgenic	WT (AB), *Tg(lyz:DsRed)*, *Tg(hsp70:p210^BCR/ABL1^)*	[105]
	ABL inhibitor imatinib, MEK inhibitor U0126, cytarabine, azacitidine, and arsenic trioxide	K562, CD34+ HPSC, MV4-11 and MOLM-13 cell xenograft	*prkdc^−/−^* in *casper* background (SCID zebrafish)	[106]
Intestine	*YES1*, *YAP1*	Morpholino-based knockdown, dasatinib treatment	WT, *Tg(fabp2:RFP)as200*, *axin1^tm213^*	[107]
	*CATSPERE*	Transgenic	WT (AB), *Tg(ifabp:DsRed-P2A-CATSPERE; CATSPERE*), *Tg(ifabp:EGFP;WT)*, *p53^−/−^*	[108]
	HCT116, anandamide	Xenograft	WT (Tübingen), *Tg(fli1:EGFP)*, *Tg(mpeg1:EGFP)*	[109]
Renal cell carcinoma	*VHL*	Transgenic	*Tg(ATPase1.a1A4:GFP)*, *Tg(vhl^hu2117+/−^)*, *Tg(ATPase1.a1A4:GFP:vhl^−/−^)*	[110]
Liver	*Tulp3*	*CRISPR* knockout	WT (AB, TL), *Tg(wt1b:EGFP)*, homozygous mutant-*tulp^m/m^*	[74]
	*Pten* and *Tp53*	*CRISPR* knockout	WT (AB), *Tg(fabp10:Cas9-mCherry);ptena^−/−^*, *Tg(fabp10:Cas9-mCherry)*; *ptenb^−/−^*, *Tg(fabp10:Cas9-mCherry)*; *tp53^−/−^*, *Tg(fabp10:Cas9-mCherry)*	[111]
Melanoma	*BRAF*	Transgenic	*WT (AB)*, *Tg(crestin:CreERt2;crystallin:YFP)*, *Tg(−3.5ubi:loxP-GFP-loxP-mCherry)*, *Tg(p53/BRAF/Na/MiniCoopR/crestin:EGFP)*	[112]
	92.1 and Mel270 cell line	Xenograft	*Tg(fli1:eGFP)*	[113]
	*MITF*, *BRAF*	Transgenic	*mitfa^vc7^*, *tp53^M214K^*, *Tg(mitfa-BRAF^V600E^)*, *Tg(mitfa:BRAF^V600E^)*; *mitfa^vc7^*; *p53^M214K^)*	[114]
	*kita* promoter, *HRAS*	Transgenic	WT (AB), *tg(UAS:GFP)*, *tg(5XUAS:eGFP-HRASV12)io6*, *mitfa^w2/w2^*, *p53^zdf1/zdf1^*, *tg(mitfa:Gal4VP16;UAS:mCherry)*; *Et(kita:GalTA4*,*UAS:mCherry)hzm1*	[115]
Myelodysplastic syndrome	*TET2*	Transgenic	WT (AB), *tet2^m/m^*, *Tg(c-myb-GFP)*, *Tg(cd41-GFP)*	[116]
	*c-myb*	Transgenic	WT (AB), *Tg(c-myb:gfp)*, *Tg(c-myb^hkz3^)*, *Tg(rag2:dsRed)*, *Tg(lyz:dsRed)*	[117]
Pancreatic cancer	*Rabl3*, *KRAS*	*CRISPR* transgenic	WT, *tp53^−/−^*, *Tg(tp53^−/−^;rabl3-TR^52^)*, *Tg(rabl3-TR^41/+^)*, *Tg(rabl3-TR^41/41^)*	[118]
	*KRAS*	Transgenic	*Tg(ubb:Lox-NucmCherry-stop-Lox-GFP::KRAS^G12D^)*, *Tg(elastase3I:CRE;cryaa:Venus)*, *Tg(ela3I-CRE*; *LSL-KRAS^G12D^)*	[119]
	Panc-1 cells	Xenograft	*Tg(fli1:eGFP)*, Nacre (*mitfa^−/−^*)	[120]
Peripheral nerve sheath tumor (PNST)	*lats1* and *lats2*	*CRISPR* knockout	WT (AB) *lats2*^mw87/mw87^	[121]
	*Suz12*	Morpholino-based knockdown	*nf1^a+/−^:nf1b^−/−^:p53^e7/e7^*	[122]
Retinoblastoma	*ACVR1C*, *SMAD*	Y79-GFP cell xenograft	WT (AB)	[123]
Rhabdomyosarcoma	*Pax3*	Transgenic	*Tg(pax3a:EGFP)*, *pax3a^−/−^* and *pax3b^−/−^*	[100]
	*HES3*, *her3*	CRISPR knockout	*her3* null mutants (*her3^nch1^*, *her3^nc2^*, *her3^nch3^*)	[101]
	*kRAS*	Transgenic	WT (AB), *Tg(myf5:GFP;mylz2:mCherry)*, *Tg(cdh15:GFP)*, *Tg(mylz2:mCherry)*, *Tg(cdh15:KRAS^G12D^)*, *Tg(mylz2:KRAS^G12D^)*	[124]
	*PAX3-FOXO1*, *HES3*	Transgenic	WT (AB), WIK, TL, AB/TL, *tp53^M214K^*, *Tg(BetaActin:GFP2A:PAX3FOXO1)*, *Tg(CMV:GFP2A:PAX3FOXO1)*, *Tg(ubi:GFP2A:PAX3FOXO1)*	[125]
Systemic mastocytosis	*KIT*	Transgenic	*Tg(actb2:KIT^D816V^:2AeGFP)*	[126]
T-cell acute lymphoid leukemia	*ptch1*	CRISPR knockout	*Tg(ptch1*^mutant^) *Tg (rag2-notch1a^ICD^)*	[96]
	*IL7R*	Transgenic	CG1, *Tg(rag2:RFP)*, *Tg(rag2:IL7Rmut2)*, *Tg(rag2:IL7R^mut2^-tdTomato)*	[127]
	*AURKB* and *Myc*	Transgenic	WT, *Tg(rag2:AURKB;rag2:mCherry)*, *Tg(rag2:Loxp-dsRED2-Loxp-EGFP-Myc;hsp70:Cre)*, *Tg(rag2:EGFP;rag2:Myc)*, *Tg(rag2:EGFP;rag2:Myc^S67A^)*, *Tg(rag2:EGFP;rag2:Myc;rag2:AURKB)*	[128]
	*prl3* and *Myc*	Transgenic	CG1, *Tg(rag2:GFP;rag2:Myc)*, *Tg(rag2:prl3;rag2:mCherry)*	[129]
	*NUP88/Nup214*	Morpholino-based knockdown	WT (AB, Tübingen)	[130]
	*Lrrc50*	ENU mutagenesis, Morpholino-based knockdown	WT, *lrrc50^hu255h (+/−)^*	[131]
Thyroid cancer	*BRAF*	Transgenic	*Tg(TdTomato-pA)*,*Tg(BRAF^V600E^-pA:TdTomato-pA)*	[132]
	*CREB3L1*	8505C cell xenograft	WT	[133]

**Table 3 cancers-16-01361-t003:** Childhood CNS cancer models in zebrafish.

Cancer	Studied Gene/Cell Line/Drug	Method	Zebrafish Line Used/Developed	Reference
Glioblastoma	*nf1*	Transgenic	*Tg(nf1a^+/–^*; *nf1b^–/–^*; *p53^e7/e7^)*	[165]
	*MTH1*, TH588 and TH1579 (MTH1 inhibitors)	GBM #18-CMV-LUC cell xenograft	Wild-type (TL)	[177]
	Microglial response toward GBM cells	U87 and U251 cell xenograft	*Tg(mpeg1:EGFP)*, *irf8^−/−^*	[176]
	Role of *tert* in telomere stability	Transgenic	*Tg(10xUAS:tert)*, *Tg(10xUAS:terc)*,	[181]
	Cxcr4-mediated infiltration of pro-tumoral macrophages	Transgenic	*Tg(NBT:∆LexPR-lexOP-pA*; *mpeg1:EGFP)*, *Tg(mpeg1:mCherry*; *p2ry12:p2ry12-GFP)*, *cxcr4b^−/−^*	[167]
Embryonal tumors (previously classified as CNS-PNETs)	*rb1*, *rbbp4*, and *hdac1*	Transgenic	*Tg(H2A.F/Z-GFP)*, *rb1^Δ7/Δ7^*	[172]
	*SOX10* and *OLIG2*	*Transgenic*, *Tumor allograft from Tg(mitfa^w2^*; *p53^M214K^*; *Tg(sox10:mCherry-NRAS^WT^) fish into mitf^w2^* fish	*Tg(mitfa^w2^*; *p53^M214K^*; *Tg(sox10:mCherry-NRAS^WT^)*, *mitf^w2^*	[173]
Medulloblastoma	CD133	Daoy cell xenograft	*Tg(flk:mCherry); Absolut^+/+^* (*ednrbl^−/−^* *mitfa^−/−^*)	[182]
Pilocytic astrocytoma	*NF1*	JHH-NF1-PA1 cell xenograft	Wild-type (AB)	[183]
Rhabdoid tumor	*SMARCB*, *PRKCD*, *DDR2*	INF_R_1288_r1, INF_R_1467_r1, and INF_R_359_r3 cell xenograft	Wild-type (AB)	[184]

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
