# Peer review of "Zebrafish—A Suitable Model for Rapid Translation of Effective Therapies for Pediatric Cancers"

_cancers, 2024, doi:10.3390/cancers16071361_

Round 1

Reviewer 1 Report (Previous Reviewer 1)

Comments and Suggestions for Authors

The review has improved a lot and it is a good reference for any person interested in pediatric cancers interesting in getting started with zebrafish

a few details:

Line 71  culture adaptation deserves a mention

Table:  zebrafish costs should be intermediate…drosophila and c elegans can be considered low

Line 171  Haffter 

Line 204 BAG or RAG ?

Line 405 “may prevent entry”  is better, depends on precise definition of “high mw”

Author Response

Response to Reviewer 1 Comments

1. Summary

Thank you very much for taking the time to review this manuscript. The comments have been very thorough and useful in improving the manuscript. We strongly believe that the comments and suggestions have increased the scientific value of revised manuscript by many folds. We have taken them fully into account in revision. Please find the detailed responses below and the corresponding revisions/corrections highlighted in the revised manuscript.

2. Point-by-point response to Comments and Suggestions for Authors

Comments and Suggestions for Authors: The review has improved a lot and it is a good reference for any person interested in pediatric cancers interesting in getting started with zebrafish.

Response: We extend our sincere gratitude to the reviewer for their insightful comment. Additionally, we would like to express our appreciation to Reviewer-1 for their meticulous inspection of the manuscript, identifying our mistakes, and providing valuable suggestions to enhance its quality.

Comments 1: Line 71 culture adaptation deserves a mention

Response 1: We agree with this comment. We have mentioned ‘Culture adaptation’ with proper references. This addition has been highlighted in ‘green’ within the main text.   

Comments 2: zebrafish costs should be intermediate…Drosophila and C. elegans can be considered low

Response 2: We agree with the comment. We have changed the table according to the reviewer’s suggestion (highlighted in ‘green’).

Comments 3: Line 171 Haffter

Response 3: We acknowledge this typing error and thank the reviewer to bring this mistake into our notice. We have rectified the mistake (highlighted in ‘green’).

Comments 4: Line 204 BAG or RAG ?

Response 4: We appreciate the reviewer for bringing the typing error to our attention and allowing us to rectify it. We have corrected the mistake by changing it to BAG1 (highlighted in ‘green’).

Comments 5: Line 405 “may prevent entry” is better, depends on precise definition of “high mw”

Response 5: We have taken reviewer’s comment in full consideration. Based on the suggestions, we have rewritten the sentence “In the early stages of development, the protective chorion of zebrafish embryos may prevent the entry of compounds that have molecular weights more than 4000 Da.” We have also provided reference for this statement.  

Reviewer 2 Report (Previous Reviewer 2)

Comments and Suggestions for Authors

accept

Comments on the Quality of English Language

none

Author Response

We would like to express our heartfelt gratitude for your invaluable contribution to our manuscript, Zebrafish- a suitable model for rapid translation of effective therapies for pediatric cancers, and for your decision to accept it for publication. Your thorough review and constructive criticism have been instrumental in shaping the final version of the manuscript. We truly appreciate the time and effort you dedicated to carefully evaluating our manuscript and providing meaningful suggestions for its enhancement.

This manuscript is a resubmission of an earlier submission. The following is a list of the peer review reports and author responses from that submission.

Round 1

Reviewer 1 Report

Comments and Suggestions for Authors

Overall this review gives a broad overview of the use of zebrafish, however in my opinion it does not do justice enough to the title. There are (too?) many reviews discussing the uses of zebrafish, and I feel there is too much text devoted to a rather generic discussion of fish and insufficient to a detailed discussion of papers that deal with childhood tumours, this is where this review could really add something new. A table is given but I think more of these papers merit a deeper discussion. For instance, are there no further good paper dealing with xenotransplants that are worth discussing in more detail?? 

Detailed comments

line 53 I would add  a “i)” before “initial tumor cells”…the term is also somewhat unclear, do you mean cell lines derived from tumour cells straight from patients? would be good to clarify

line 60 “this model” what is meant by this …the in vitro model? …better to be explicit

line 72-77 say something about  immune and other cell types in these organoids??

line 89 metabolism

the phrasing in the disadvantages box for zebrafish is odd: the difference in temperature and different means of drug exposure (adding to external environment, uptake issues) could could reduce cilinical relevance of the obtained results.

Larvae lack an adaptive immune system

Line 147 need references for the diseases mentioned 

Line 148-165 does not seem very relevant for this review and could be shortened

Table 1 is called cancer models..however, when I looked up one of the references it was simply a gene knockdown without presence of any cancer formation in zebrafish, I would either restrict the table to real “cancer models” in zebrafish or rename the table in something more generic.

The group of M. Mione has done work specifically on pedriatic brain tumors in fish but was not cited..I think this needs checking out and including

Line 182-212 forms a rough description of zebrafish-human brain similarity  but it is not really clear why this needs to be in this review, as it is on pediatric cancer…I think it needs something at the start that makes clear why this text is relevant for  a review on pediatric tumours.

The statement on line 220 on transgenics feels a bit bland unless 1-2 strong examples are given… the rb example that follows: how does it relate two the starting sentences of the paragraph?  I would rather expect a xenotransplant  example.

Overall this paragraph feels disjointed.

CNS-PNETs are not really properly introduced…is this a childhood cancer? if yes give some intro on this and characteristics before going into the zebrafish model

line 232 I would state “pediatric brain tumor”

line 233-235 larval lethal…did it have tumours? If so I would state explicitly, in what way was the model close to human pathological conditions and severity?

line 236 “triple mutant heterozygous nf1”  is wrong and confusing why not simply state their genotype? ..ie in nf1a+/–; nf1b–/–; p53e7/e7 fish

I would like to see more specific models discussed, not just a table, I have a feeling there should be more papers

Lines 259-265 are irrelevant, rather discuss more pediatric tumour papers

Line 307 Immunodeficient zebrafish are mentioned but not referenced clearly, humanised zebrafish has a separate paragraph.

line 311  It is likely that the skin is also a major barrier

Comments on the Quality of English Language

good

Reviewer 2 Report

Comments and Suggestions for Authors

The review is very interesting and overall well written.

Overall, the topic of this review is of relevance for the scientific community and I think worth being published. However, the manuscript in its current form appears rather preliminary and not really carefully crafted, resembling more a "draft" than a final version.

A couple of recent reviews on the same topic have been recently published:

Pediatric Cancer Models in Zebrafish.

Casey MJ, Stewart RA. Trends Cancer. 2020 May;6(5):407-418. doi: 10.1016/j.trecan.2020.02.006.

Zebrafish Models of Paediatric Brain Tumours.

Basheer F, Dhar P, Samarasinghe RM. Int J Mol Sci. 2022 Aug 31;23(17):9920. doi: 10.3390/ijms23179920.

The following pertinent reports should be mentioned/discussed:

A map of cis-regulatory elements and 3D genome structures in zebrafish.

Yang H, Luan Y, Liu T, Lee HJ, Fang L, Wang Y, Wang X, Zhang B, Jin Q, Ang KC, Xing X, Wang J, Xu J, Song F, Sriranga I, Khunsriraksakul C, Salameh T, Li D, Choudhary MNK, Topczewski J, Wang K, Gerhard GS, Hardison RC, Wang T, Cheng KC, Yue F. Nature. 2020 Dec;588(7837):337-343. doi: 10.1038/s41586-020-2962-9.

Visualizing Engrafted Human Cancer and Therapy Responses in Immunodeficient Zebrafish.

Yan C, Brunson DC, Tang Q, Do D, Iftimia NA, Moore JC, Hayes MN, Welker AM, Garcia EG, Dubash TD, Hong X, Drapkin BJ, Myers DT, Phat S, Volorio A, Marvin DL, Ligorio M, Dershowitz L, McCarthy KM, Karabacak MN, Fletcher JA, Sgroi DC, Iafrate JA, Maheswaran S, Dyson NJ, Haber DA, Rawls JF, Langenau DM. Cell. 2019 Jun 13;177(7):1903-1914.e14. doi: 10.1016/j.cell.2019.04.004.

Zebrafish Cancer Predisposition Models.

Kobar K, Collett K, Prykhozhij SV, Berman JN. Front Cell Dev Biol. 2021 Apr 27;9:660069. doi: 10.3389/fcell.2021.660069

Zebrafish Cancer Avatars: A Translational Platform for Analyzing Tumor Heterogeneity and Predicting Patient Outcomes.

Al-Hamaly MA, Turner LT, Rivera-Martinez A, Rodriguez A, Blackburn JS. Int J Mol Sci. 2023 Jan 24;24(3):2288. doi: 10.3390/ijms24032288.

Zebrafish her3 knockout impacts developmental and cancer-related gene signatures.

Kent MR, Calderon D, Silvius KM, Kucinski JP, LaVigne CA, Cannon MV, Kendall GC. Dev Biol. 2023 Apr;496:1-14. doi: 10.1016/j.ydbio.2023.01.003

The presentation and critical interpretation of results of previous studies should be improved.

The quality of the figures should be improved, including informative aspects for the Readers; professional assistance should be sought.

Comments on the Quality of English Language
